# Hepatitis C Infection as a Risk Factor for Hypertension and Cardiovascular Diseases: An EpiTer Multicenter Study

**DOI:** 10.3390/jcm11175193

**Published:** 2022-09-01

**Authors:** Paweł Rajewski, Dorota Zarębska-Michaluk, Ewa Janczewska, Andrzej Gietka, Włodzimierz Mazur, Magdalena Tudrujek-Zdunek, Krzysztof Tomasiewicz, Teresa Belica-Wdowik, Barbara Baka-Ćwierz, Dorota Dybowska, Waldemar Halota, Beata Lorenc, Marek Sitko, Aleksander Garlicki, Hanna Berak, Andrzej Horban, Iwona Orłowska, Krzysztof Simon, Łukasz Socha, Marta Wawrzynowicz-Syczewska, Jerzy Jaroszewicz, Zbigniew Deroń, Agnieszka Czauż-Andrzejuk, Jolanta Citko, Rafał Krygier, Anna Piekarska, Łukasz Laurans, Witold Dobracki, Jolanta Białkowska, Olga Tronina, Magdalena Wietlicka-Piszcz, Małgorzata Pawłowska, Robert Flisiak

**Affiliations:** 1Department of Internal and Infectious Diseases, Provincial Infectious Disease Hospital, 85-030 Bydgoszcz, Poland; 2Department of Infectious Diseases, Voivodship Hospital and Jan Kochanowski University, 25-369 Kielce, Poland; 3Hepatology Outpatient Clinic, ID Clinic, 41-400 Mysłowice, Poland; 4Department of Internal Medicine and Hepatology, Central Clinical Hospital of the Ministry of Internal Affairs and Administration, 02-507 Warsaw, Poland; 5Clinical Department of Infectious Diseases, Specialist Hospital in Chorzów, Medical University of Silesia, 40-055 Katowice, Poland; 6Department of Infectious Diseases and Hepatology, Medical University of Lublin, 20-059 Lublin, Poland; 7Regional Center for Diagnosis and Treatment of Viral Hepatitis and Hepatology, John Paul II Hospital, 31-202 Kraków, Poland; 8Department of Infectious Diseases and Hepatology, Faculty of Medicine, Collegium Medicum Bydgoszcz, Nicolaus Copernicus University, 87-100 Toruń, Poland; 9Pomeranian Center of Infectious Diseases, Department of Infectious Diseases, Medical University of Gdańsk, 80-210 Gdańsk, Poland; 10Department of Infectious and Tropical Diseases, Jagiellonian University Collegium Medicum, 30-252 Kraków, Poland; 11Hospital for Infectious Diseases in Warsaw, 01-201 Warsaw, Poland; 12Department of Infectious Diseases and Hepatology, Wrocław Medical University, 50-367 Wrocław, Poland; 13Department of Infectious Diseases, Hepatology and Liver Transplantation, Pomeranian Medical University, 70-204 Szczecin, Poland; 14Department of Infectious Diseases, Medical University of Silesia in Katowice, 41-902 Bytom, Poland; 15Ward of Infectious Diseases and Hepatology, Biegański Regional Specialist Hospital, 91-347 Łódź, Poland; 16Department of Infectious Diseases and Hepatology, Medical University of Białystok, 15-089 Białystok, Poland; 17Medical Practice of Infections, Regional Hospital, 10-561 Olsztyn, Poland; 18Infectious Diseases and Hepatology Outpatient Clinic NZOZ “Gemini”, 62-571 Żychlin, Poland; 19Department of Infectious Diseases and Hepatology, Medical University of Łódź, 90-419 Łódź, Poland; 20Multidisciplinary Regional Hospital in Gorzów Wielkopolski, 66-400 Gorzów Wielkopolski, Poland; 21MED-FIX Medical Center, 53-522 Wrocław, Poland; 22Department of Infectious and Liver Diseases, Medical University of Łódź, 90-419 Łódź, Poland; 23Department of Transplantation Medicine, Nephrology, and Internal Diseases, Medical University of Warsaw, 02-091 Warsaw, Poland; 24Department of Theoretical Fundations of Biomedical Sciences and Medical Informatics, Nicolaus Copernicus University, 87-100 Toruń, Poland

**Keywords:** hepatitis C, HCV genotype, hypertension, cardiovascular diseases

## Abstract

Hepatitis C infection is one of the main reasons for liver cirrhosis and hepatocellular carcinoma. In recent years, more and more is being heard about extrahepatic manifestations of the hepatitis C infection including its possible influence on the development of hypertension and cardiovascular diseases. In the given work, the frequency analysis of the incidence of hypertension and cardiovascular diseases among 2898 HCV-infected patients treated in Poland and the assessment of their relevance to the HCV genotype and the progression of liver fibrosis can be found. The prevalence of hypertension in the group of analyzed patients was 39% and was significantly associated with old age (OR = 1.08 (1.07–1.08)) and female sex, as well as the progression of liver fibrosis (OR = 1.54 (1.29–1.85)). Hypertension was found in 47.6% of patients with F4 fibrosis, 42.1% of patients with F3 fibrosis, and 25% of patients with F1 fibrosis. The incidence of cardiovascular disease in the studied group of patients was as follows: all incidents, 131 (4.52%); including ischemic heart disease 104, (3.95%); stroke, 2 (0.07%); atherosclerosis, 21 (0.72%); and aneurysms, 4 (0.14%). The obtained results prove that the prevalence of cardiovascular diseases is significantly associated with the advanced age of patients and the progression of liver fibrosis. The relevance of sex and the HCV genotype to the prevalence frequency of cardiovascular diseases in the study group has not been proven. This being the case, no differences in the frequency of their incidence depending on the HCV genotype, including genotype 3, was found. Hepatitis C infection as a non-classical risk factor for cardiovascular disease and hypertension does require further studying.

## 1. Introduction

Hepatitis C infection is one of the main reasons for chronic liver diseases, cirrhosis, and liver cancer in the world contributing to a decrease in life quality and longevity [1,2,3,4].

In Poland, about 80% of infections are found with HCV genotype 1b, whereas 14% are with genotype 3, and 5% of the infected patients are with genotype 4 [5].

The dangers associated with the infection result from: a huge spread of the virus in the population; few symptoms long-term, or even an asymptomatic course of the disease; extrahepatic manifestation of the virus; its low detection rate; and lack of vaccination [3].

In recent years, studies have proved that hepatitis C infection also contributes to the development of metabolic disorders which play a crucial role as risk factors for cardiovascular disease and hypertension. A noticeable influence of HCV on the development of obesity, insulin resistance, diabetes development, lipid disturbances, and hepatic steatosis resulted in the fact that metabolic disturbances in the course of the infection are named ‘metabolic and virus syndrome’ by some authors, whereas hepatic steatosis is described as an organ form of a metabolic syndrome [6,7,8,9,10,11]. The observed increased presence of diseases of the heart and vessels among patients suffering from chronic hepatitis C virus contributed to the indication of HCV as a new risk factor for cardiovascular diseases and its presence as an extrahepatic manifestation of the hepatitis C infection.

Cardiovascular diseases, including ischemic heart disease, cerebrovascular diseases, and peripheral vascular diseases, are the reason for about 4.3 million of deaths in Europe annually which equates to nearly half of all deaths at 48% (54% of women and 43% of men). In Poland, the numbers equate to 46% of deaths annually (51.1% of women and 40.9% of men). The incidence frequency of ischemic heart disease increases with the increase in age, from 5–7% among women within the age range 45–64 to 10–12% among women within the age range 65–84, and 4–7% among men within the age range 45–64 to 12–14% among men within the age range 65–84 [12].

The main risk factors for the development of cardiovascular diseases in Poland are hypertension, lipid disturbances, obesity, diabetes, and smoking. New risk factors seem to be infectious agents, for example, the influenza virus and human immunodeficiency virus (HIV). In previous years, the infection of HCV seemed to be indicated as a possible developmental reason for cardiovascular diseases [13,14].

The potential role of HCV as a risk factor for the development of cardiovascular diseases seems to be quite complex. On the one hand, the infection leads in a direct way to chronic inflammation, contributing to the development of arteriosclerosis and function disturbances of the vascular endothelium; on the other hand, hepatitis C infection leads in an indirect way to the development of other key risk factors such as diabetes, obesity, hypertriglyceridaemia, hypertension, and chronic kidney diseases [8,9,10,11].

The mentioned metabolic disturbances with regard to HCV are similar to those which are observed in the so-called metabolic syndrome (obesity, hypertension, diabetes and incorrect blood glucose level and glucose intolerance, increased concentration of triglycerides, and decreased concentration of HDL cholesterol in blood serum) which is thought nowadays to be the main reason for the increased risk of cardiovascular diseases.

The most vital seem to be carbohydrate disorders which result in the development and increase in atherosclerosis of the coronary arteries as well as brain arteries and peripheral ones.

The incidence of hypertension, one of the most frequent risk factors for cardiovascular diseases, amounts to 24.1% among men and 20.1% among women. In Poland, according to NATPOL 2011, the prevalence of hypertension among persons between the ages of 18–79 increased from 30 to 32% in the last 10 years, which equates to about nine million persons [12].

## 2. The Aim of the Work

The aim of the work is the frequency assessment of the prevalence of cardiovascular diseases and hypertension of persons suffering from HCV and an attempt to assess their relevance to the progression of liver fibrosis and the HCV genotype.

## 3. Material and Method

The studies covered 2898 patients with chronic HCV, including 1486 women (51%) and 1412 men (49%) with an age range from 19 to 91 (median age 58), who were qualified for antiviral treatment based on the EpiTer multicenter study database (Table 1).

The analysis of medical documentation was carried out, without physical examination, in terms of the incidence and prevalence of cardiovascular diseases and hypertension. HCV genotype was defined for each patient, and liver fibrosis was assessed by means of liver biopsy or elastography. In the analyzed work, the definition of HCV genotype 1 includes genotype 1 with subtypes 1a and 1b.

Inclusion criteria: each patient meeting the criteria for the treatment of chronic hepatitis C under the Drug Program of the National Health Fund in Poland was included in the study. Patients over 18 years of age, diagnosed with HCV RNA over at least a 6 month-duration and with at least F1 liver fibrosis according to Metavir or extrahepatic HCV manifestations, regardless of the presence of liver fibrosis, were eligible for treatment. 

Exclusion criteria: age under 18, pregnant or breastfeeding, no confirmed HCV RNA in blood serum, no evidence of fibrosis in liver biopsy or liver elastography, no patient consent.

The presence of cardiovascular diseases and hypertension was assessed on the basis of the retrospective analysis of the medical records of the patients. Cardiovascular diseases that were taken into account in the presented study are ischemic heart disease in the form of: chronic coronary syndrome (stable angina pectoris), previous acute coronary syndrome (myocardial infarction), previous coronary angioplasty (PTCA), past coronary artery bypass grafting (CABG), previous ischemic stroke, previous transient ischemic attack (TIA), carotid atherosclerosis, lower limb atherosclerosis, previous lower limb artery angioplasty, and aortic aneurysm. 

Liver fibrosis was assessed on the basis of classic core needle biopsy or liver elastography performed with the FibroScan Touch 502 device. The histopathological results of liver biopsies from up to 5 years ago were taken into account, and the results of liver elastography from up to 1 year ago. 

The determination of HCV RNA and HCV genotype was performed in standardized laboratories by PCR and nucleic acid hybridization methods. 

Patients were treated with direct-acting antiviral therapy (DAA) under the Drug Program of the National Health Fund in Poland. The type and duration of treatment depended on the HCV genotype, stage of fibrosis, history of prior treatment, and drug interactions. 

Treatment was performed using: sofosbuvir, ledipasvir; sofosbuvir, ribavirin; elbasvir, grazoprevir; ombitasvir, paritaprevir, ritonavir, dasabuvir; ombitasvir, paritaprevir, ritonavir; and daclatasvir, asunaprevir. Treatment was combined with ribavirin in certain cases.

The summary statistics for continuous, non-normally distributed variables are presented as median with range; categorical variables are presented as frequencies. Differences between continuous variables were analyzed using the Wilcoxon test. Differences for categorical variables were assessed using the chi-square or Fisher exact test for independence. The assessment of the factors potentially associated with hypertension and heart diseases was carried out using univariable and multivariable logistic regression (LR). Variables significant at the 0.2 level in the univariable models were considered for inclusion in the multivariable model. The results were considered as statistically significant when the *p*-value was lower than 0.05. The statistical analysis was performed with the use of R software, version 3.0.3.

## 4. Results

The prevalence of hypertension in the analyzed patients group is significantly associated with the old age of patients and female sex as well as the progression of liver fibrosis.

Further analysis was carried out by means of logistic regression in order to determine the independent factors related to the prevalence of hypertension in the study group.

The results obtained by means of logistic regression prove that more frequent hypertension prevalence was related to the progression of fibrosis, OR = 1.54 (1.29–1.85) and old age, OR = 1.08 (1.07–1.08). Patients infected by HCV genotype 1 or 4 suffered from hypertension more often than patients with HCV genotype 3.

The incidence of cardiovascular disease in the studied group of patients was as follows: all incidents, 131 (4.52%); including ischemic heart disease, 104 (3.95%); stroke, 2 (0.07%); atherosclerosis, 21 (0.72%); and aneurysms, 4 (0.14%).

The obtained results prove that the prevalence of cardiovascular diseases is significantly associated with the advanced age of patients and the progression of liver fibrosis. The relevance of sex and the HCV genotype to the prevalence frequency of cardiovascular diseases in the study group has not been proven.

Further analysis was carried out by means of logistic regression in order to determine the independent factors related to the incidence of cardiovascular diseases in the study group.

## 5. Discussion

In previous years, studies have proved the relevance of chronic hepatitis C infection to the increase in risk of cardiovascular diseases, their complications, and the increase in deaths (1.65 times) in relation to the population of noninfected persons. The increase in risk of cardiovascular diseases in patients suffering from chronic hepatitis C virus with an additional risk factor such as diabetes or hypertension is 1.75 times greater in relation to noninfected persons [14].

It has been proved, on the basis of a profound observational study from Taiwan, that HCV is an independent factor for a stroke and increases the risk of the incidence of peripheral atherosclerosis by 1.43 times [15,16,17].

A decrease in the number of acute coronary syndromes and ischemic strokes has been proved, with the diseases of patients suffering from chronic hepatitis C virus and treated casually in comparison to patients who were not treated [18].

American observational studies have proved that patients with the detectable RNA HCV, suffered statistically more often from a coronary event [19].

As for personal studies of persons suffering from chronic hepatitis C, hypertension has been found in the cases of more than 39% of the examined group of persons, which equates to an increase of 7% in comparison to the general population (NATPOL 2011 studies). Furthermore, women with HCV more often suffered from hypertension (41.6%) unlike in the general population. More frequent incidence of hypertension was associated with age of a patient and the progression of fibrosis. Hypertension was found in 46.7% of patients with F4 fibrosis, 42.1% of patients with F3 fibrosis, and in 25% of patients with F1 fibrosis in the studied group (Table 2 and Table 3).

Cardiovascular diseases were found in 4.55% of patients suffering from HCV. The incidence was characterized by a similar frequency of both women and men, which constitutes a similar percentage in relation to the general population in Poland. More frequent incidence was associated with the progression of fibrosis and the advanced age of a patient (Table 4 and Table 5). The age of a patient (>45 years old for a man and >55 years old for a woman) is an independent risk factor for the incidence of hypertension and cardiovascular diseases both in a general population and in a group of patients with HCV. (Table 5). It reflects the probability of the emergence of other risk factors. Old age is a potentially longer time for the overlapping of single risk factors for cardiovascular diseases. As far as the general population is concerned, male sex is associated with a three times higher risk of coronary event and four times higher risk of death due to coronary reasons. It is associated with the presence of other risk factors for cardiovascular diseases such as: smoking, a raised level of total cholesterol, decrease in HDL cholesterol level, and the incidence of hypertension. The phenomenon of gender differences is explained with the protective role of estrogens of premenopausal women: estrogens contribute to the regulation of carbohydrate metabolism and lipid parameters, and the function of the vascular endothelium and homeostatic system. As far as women with HCV are concerned, hormonal imbalances are also found with the progression of liver fibrosis, and hormonal imbalances including the production of estrogens, which is age independent, may contribute to the incidence of hypertension in the group of patients.

There are a few factors which are associated with a higher frequency of hypertension incidence in patients with HCV in comparison to the general population. The development of inflammation seems to be one of the factors which contributes to the development of arteriosclerosis and vessel damage. Ceramides composed of long-chain saturated fatty acids contribute to the development of the inflammatory state of patients suffering from HCV. The consequence of inflammatory state is adipocytes apoptosis, macrophage mobilization, the creation of inflammatory infiltration, and the release of free oxygen radicals, TNF-alfa, FFA, and PA I 3, which contribute to the development of hepatic steatosis, insulin resistance, obesity and, as a consequence, the development of arteriosclerosis and hypertension. TNF-alpha dysregulates post-receptor pathways for insulin, blocks the connection of an insulin receptor with IRS-1, inhibits phosphatidyl-inositol-3-kinase, and dysregulates the synthesis of glycogen, fats, and proteins. IL-6 reduces autophosphorylation of the insulin receptor and IRS-2 phosphorylation. Oxidative stress also seems to contribute to metabolic disorders which results in the development of arteriosclerosis and the increase in the blood pressure of patients suffering with HCV. It may result in insulin resistance due to the activation of protein kinase, c-Jun N-terminal kinase, which results in the degradation of IRS [20,21]. The influence of HCV on the activity of cytokines produced by fatty tissue and the related increase in body mass and development of obesity may be responsible for the increased number of hypertension cases in patients with HCV. The decreased concentration of adiponectin, and leptins is observed in the group of patients as well as the increased concentration of resistin, ghrelinm, and visfatin, which contributes to inflammation, insulin resistance, lipid disorders, and arteriosclerosis, and as a consequence, to the development of hypertension [22].

Extrahepatic manifestations of HCV, such as membranous and membranoproliferative glomerulonephritis which may be the reason for hypertension in the study group but have not been analyzed in the given study, could also contribute to the received result [7]. Together with the progression of liver fibrosis, the frequency increase in hypertension was proved in the study group of patients with HCV. Similar results were received in other studies; however, as far as patients with clinic symptoms of liver cirrhosis in B and C classes according to the Child–Pugh scale are concerned, especially those with advanced portal hypertension, ascites, and hepatic failure, lower rather than higher blood pressure results may be noticed significantly more often. In the studied population of patients with HCV, hepatic fibrosis was the most often assessed by means of noninvasive methods, mainly by means of elastography of the liver using the FibroScan method, which may explain the fact that there were no patients with ascites and clinical features of hepatic cirrhosis (B and C classes according to the Child–Pugh scale) among patients with advanced fibrosis of F3 and F4 type. The majority of them were characterized by A class according to the Child–Pugh scale [22,23,24,25].

As far as decompensated cirrhosis is concerned, especially when long-lasting, the dominance of vasodilation over vasoconstriction is found as well as a decrease in blood pressure or even its normalization. The given phenomenon is observed among patients with identified long-term primary hypertension as well as secondary hypertension and resistant hypertension.

The incidence of portal hypertension and ascites of patients suffering from cirrhosis results in significant hemodynamic changes in the cardiovascular system, which as a consequence, contribute to a decrease in blood pressure rather than its increase due to the fact that the development of alternative portosystemic connections may be observed. When visceral vascular endothelium is exposed to increased blood flow and nonphysiological high blood pressure, it releases nitric oxide as a response to mechanical stimuli: nitric oxide constitutes a main factor contributing to vasodilation which results in a decrease in blood pressure and the resistance of peripheral vessels. The development of collateral circulation contributes to the increased level of vascular endothelial growth factor, which is responsible for angiogenesis and additionally, activating the endothelial synthesis of NO. Prostacycline (PG12), carbon monoxide, anandamide, vasoactive intestinal peptide, adrenomedullin, substance P, glucagon, and bile acids [26,27,28,29,30,31,32,33] play an additional role in vasodilatation.

As far as patients suffering from hepatic fibrosis are concerned, a transient increase in blood pressure may be noticed, in the first stage, related to stimulation of the sympathetic nervous system, release of endothelin 1 (ET-1), neuropeptide Y, and stimulation of the renin–angiotensin–aldosterone system (RAA), which leads to hyperaldosteronism, the retention of sodium and water and an increased level of vasopressin, which contributes to the increase in blood pressure without significant pressure increase in the portal vein and an increase in ascites. Blood pressure related to the progression of fibrosis is not only a result of functional changes in vessels but also of their remodeling caused by arteriosclerosis which additionally potentiates action of the HCV [34,35,36].

Moreover, when making the assessment of the circadian rhythm of blood pressure of patients with cirrhosis, less pressure reduction is found at night, so called non-dippers, which probably results from the dysfunction of the autonomic nervous system [37,38].

The relevance between HCV genotype and the incidence of hypertension and cardiovascular diseases has not been proved. Low number of patients with an HCV genotype other than type 1 participating in the study, resulting from the population genotype distribution found in Poland, may have contributed to the above mentioned fact.

In comparison to other research, the received dissimilar results related to the lack of increase in cardiovascular diseases among patients with HCV may result from a low percentage of patients with HCV genotype 3 (7.5%), a genotype which is closely associated with the incidence of metabolic disorders such as insulin resistance, diabetes, lipid disturbances, arteriosclerosis, and cirrhosis [10,11]. The given analysis compares: the same number of persons with HCV genotypes 1 and 3; within the age ranges <40, between the ages of 40 and 60, and >60; with the same progression of fibrosis; and similar as far as sex is concerned (Table 6 and Table 7). No relevance between genotype and hypertension and cardiovascular diseases was found.

HCV treatment with DAA affects practically all asepsis of hepatic and extrahepatic HCV manifestations [39].

The introduction of HCV treatment with DAA significantly improved the prognosis of HCV patients, not only by reducing cardiovascular risk but also by influencing other risk factors, especially metabolic disorders such as insulin resistance, diabetes, fatty liver disease, and carotid atherosclerosis. HCV eradication with DAA also aids a reduction in major adverse cardiac events (MACEs) [40,41,42]. The influence of HCV treatment with DAA on the occurrence of MACE is particularly visible in the population of patients with prediabetes, namely, insulin resistance. In this group of patients, HCV eradication by DAAs allows a significant reduction in MACEs [43]. A prospective multicenter study of 2204 HCV patients proved that the clearance of HCV by direct-acting antivirals (DAA) significantly reduced the annual incidence of cardiovascular events by 0.68%, and the CV risk was reduced by 2.0–3.5 fold [44].

The reduction in insulin resistance and diabetes mellitus events in HCV patients treated with DAA is particularly important in the context of reducing cardiovascular risk. In many recent studies, the influence of HCV on insulin resistance and the development of type 2 diabetes has been proven. For years, HCV has also been considered a risk factor for the development of diabetes, and diabetes is a manifestation of extrahepatic HCV. Therefore, HCV treatment has a beneficial effect on many cardiovascular risk factors [42,43].

HCV treatment with DAA obviously reduces the rate of liver cirrhosis, HCC development, the number of liver transplants, and metabolic disorders in the course of the disease. Recent studies have shown that the type of DAA treatment is also important, especially for the development of HCC [45,46].

A limitation of the study was, to some extent, the cardiovascular disease analysis based only on the medical records of the patients, which might not contain all the relevant data on cardiovascular disease and other relevant risk factors. In addition, although it is one of the methods recommended and used routinely in hepatology, the assessment of liver fibrosis based on liver elastography does not always reflect the actual stage of the disease. It mainly concerns advanced fibrosis at an F4 level according to Metavir, which does not always correlate with clinical cirrhosis. Furthermore, although the mean age was 58 years, the youngest patient was 19 years old and the age distribution of the patients could have influenced the results obtained. With age, the risk of cardiovascular risk factors, cardiovascular disease, and hypertension increases.

HCV infection leads to many disorders and extrahepatic manifestations, which increase the risk of cardiovascular diseases, especially insulin resistance, diabetes, fatty liver or atherosclerosis. It has been shown that eradication of HCV infection can reduce this risk, improving the quality and life expectancy of patients. Early detection of HCV infection and appropriate DAA treatment is an important factor in reducing this risk. Therefore, people infected with HCV should be actively searched for by conducting educational and screening campaigns for the presence of anti-HCV antibodies.

## 6. Conclusions

Hepatitis C infection seems to be a risk factor for hypertension.

The progression of liver fibrosis is associated with more frequent incidences of hypertension among patients with HCV.

The age of the patient and the progression of liver fibrosis are associated with more frequent incidences of cardiovascular diseases among patients with HCV.

The HCV genotype does not affect the frequency of incidence of hypertension and cardiovascular diseases.

HCV infection may be considered a non-classical risk factor for cardiovascular diseases.

## Figures and Tables

**Table 1 jcm-11-05193-t001:** The characteristics of the study group.

Total Number		*n* = 2898
		*n* (%)
**Sex**		
	female	1486 (51.28)
	male	1412 (48.72)
**Genotype**		
	1	2596 (89,58)
	3	219 (7.56)
	4	83 (2.86)
**Fibrosis (F) ****		
	1	624 (22.74)
	2	391 (14.25)
	3	462 (16.84)
	4	1267 (46.17)
**Age (years)**		58 (19–91) *

* Median (Range); ** Liver fibrosis was not determined for 154 persons.

**Table 2 jcm-11-05193-t002:** The characteristics of patients—hypertension—in relation to all the study group.

		Hypertension	
		*n* = 1132	
		*n* (%)	*p*
Sex			
	female	618 (41.6)	
	male	514 (36.4)	0.004
**Genotype**			
	1	1029 (39.6)	
	3	69 (31.5)	
	4	34 (41)	0.057
Fibrosis			
	0–2	286 (28.2)	
	3–4	797 (46.1)	<0.001
**Age (years)**		62 (22–90) *	<0.001

* Median (Range).

**Table 3 jcm-11-05193-t003:** Factors associated with the prevalence of hypertension—logistic regression.

	Simple LR		Multiple LR	
	OR (CI)	*p*	OR (CI)	*p*
Sex M vs. F	0.8 (0.69–0.93)	0.0043	1.13 (0.95–1.34)	0.1781
Age (years)	1.08 (1.07–1.09)	<0.001	1.08 (1.07–1.08)	<0.001
Fibrosis 3–4 vs. 0–2	2.18 (1.85–2.57)	<0.001	1.54 (1.29–1.85)	<0.001
Genotype 1 vs. 3	1.43 (1.06–1.92)	0.0183	1.29 (0.93–1.78)	0.1243
Genotype 4 vs. 3	1.51 (0.89–2.54)	0.1228	2.28 (1.26–4.11)	0.0065

OR (CI): odds ratio with 95% confidence interval; SE: standard error of the estimate.

**Table 4 jcm-11-05193-t004:** The characteristics of patients—cardiovascular diseases—in relation to the all study group.

		Cardiovascular Diseases	
		*n* = 131	
		*n* (%)	*p*
Sex			
	female	67 (4.5)	
	male	64 (4.5)	1
**Genotype**			
	1	118 (4.5)	
	3	10 (4.6)	
	4	3 (3.6)	0.921
Fibrosis			
	1–2	32 (3.2)	
	3–4	92 (5.3)	0.008
**Age (years)**		66.5 (39–84) *	<0.001

* Median (Range).

**Table 5 jcm-11-05193-t005:** Factors related to the incidence of cardiovascular diseases—logistic regression.

	Simple LR				Multiple LR			
	Estimate	SE	OR (CI)	*p*	Estimate	SE	OR (CI)	*p*
Fibrosis 3–4 vs. 0–2	0.5589	0.2089	1.75 (1.16–2.63)	0.0075	0.1619	0.2148	1.18 (0.77–1.79)	0.4511
Age	0.0835	0.009	1.09 (1.07–1.11)	<0.001	0.0838	0.0094	1.09 (1.07–1.11)	

Only the advanced age of a patient related to more frequent incidence of cardiovascular diseases, OR = 1.09 (1.07–1.11), in the analysis by means of logistic regression.

**Table 6 jcm-11-05193-t006:** The analysis of the incidence of hypertension.

**Group of patients < 40 years old:**				
		**Overall**	**Hypertension**	
		n (%)	n (%)	*p*
**Sex**				
	female	214 (41.31)	21 (9.8)	
	male	304 (58.69)	47 (15.5)	0.065
**Fibrosis**				
	1	208 (42.11)	14 (6.7)	
	2	88 (17.81)	13 (14.8)	
	3	67 (13.56)	6 (9)	
	4	131 (26.52)	34 (26)	<0.001
**Fibrosis**				
	0–2	296 (59.92)	27 (9.1)	
	3–4	198 (40.08)	40 (20.2)	0.001
**Genotype N**				
	1	461 (89)	58 (12.6)	
	3	36 (6.95)	6 (16.7)	
	4	21 (4.05)	4 (19)	0.567
**Age (years)**		35 (19–40) *	39 (22–40)	<0.001
**Group of patients > 60 years old:**				
		**Overall**	**Hypertension**	
		n (%)	n (%)	*p*
**Sex**				
	female	708 (62.71)	427 (60.3)	
	male	421 (37.29)	240 (57)	0.288
**Fibrosis**				
	1	154 (14.53)	80 (51.9)	
	2	130 (12.26)	66 (50.8)	
	3	184 (17.36)	117 (63.6)	
	4	592 (55.85)	374 (63.2)	0.007
**Fibrosis**				
	0–2	284 (26.79)	146 (51.4)	
	3–4	776 (73.21)	491 (63.3)	0.001
**Genotype N**				
	1	1057 (93.62)	618 (58.5)	
	3	57 (5.05)	36 (63.2)	
	4	15 (1.33)	13 (86.7)	0.071
**Age (years)**		66 (61–91)	67 (61–90)	<0.001

* Median (Range).

**Table 7 jcm-11-05193-t007:** Logistic regression in the analyzed age groups.

	Simple LR				Multiple LR			
Age <= 40	Estimate	SE	OR (CI)	*p*	Estimate	SE	OR (CI)	*p*
Sex M vs. F	0.5192	0.2792	1.68 (0.97–2.91)	0.0629	0.3344	0.2957	1.4 (0.78–2.49)	0.2581
Age (years)	0.1441	0.0329	1.15 (1.08–1.23)	0	0.1325	0.0333	1.14 (1.07–1.22)	0.0001
Fibrosis 3–4 vs. 0–2	0.9252	0.2685	2.52 (1.49–4.27)	0.0006	0.6277	0.284	1.87 (1.07–3.27)	0.0271
Genotype 1 vs. 3	−0.3291	0.4687	0.72 (0.29–1.8)	0.4827	0.0195	0.4857	1.02 (0.39–2.64)	0.968
Genotype 4 vs. 3	0.1625	0.7133	1.18 (0.29–4.76)	0.8198	0.7893	0.7626	2.2 (0.49–9,81)	0.3007
Age 40–60								
Sex M vs. F	0.1343	0.1225	1.14 (0.9–1.45)	0.2729	0.2459	0.1318	1.28 (0.99–1.66)	0.0622
Age (years)	0.0754	0.0116	1.08 (1.05–1.1)	0	0.0755	0.0122	1.08 (1.05–1.1)	0
Fibrosis 3–4 vs. 0–2	0.4383	0.1333	1.55 (1.19–2.01)	0.001	0.341	0.1384	1.41 (1.07–1.84)	0.0138
Genotype 1 vs. 3	0.5796	0.2266	1.79 (1.15–2.78)	0.0105	0.6398	0.2359	1.9 (1.19–3.01)	0.0067
Genotype 4 vs. 3	0.7313	0.3732	2.08 (1–4.32)	0.0501	0.9256	0.4004	2.52 (1.15–5.53)	0.0208
Age >60								
Sex M vs. F	−0.1363	0.1249	0.87 (0.68–1.11)	0.2751	−0.0515	0.1344	0.95 (0.73–1.24)	0.7014
Age (years)	0.0798	0.012	1.08 (1.06–1.11)	0	0.0791	0.0126	1.08 (1.06–1.11)	0
Fibrosis 3–4 vs. 0–2	0.4876	0.1401	1.63 (1.24–2.14)	0.0005	0.4641	0.1434	1.59 (1.2–2.11)	0.0012
Genotype 1 vs. 3	−0.197	0.2816	0.82 (0.47–1.43)	0.4842	−0.3375	0.2981	0.71 (0.4–1.28)	0.2576
Genotype 4 vs. 3	1.3328	0.8077	3.79 (0.78–18,46)	0.0989	1.1807	0.8224	3.26 (0.65–16.32)	0.1511

Factors associated with the incidence of hypertension and results of the logistic regression analysis; OR (CI): odds ratio with 95% confidence interval; SE: standard error of the estimate.

## Data Availability

Data supporting reported results can be provided upon request from the corresponding author.

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
