# Peer review of "Hepatitis C Infection as a Risk Factor for Hypertension and Cardiovascular Diseases: An EpiTer Multicenter Study"

_jcm, 2022, doi:10.3390/jcm11175193_

Round 1

Reviewer 1 Report (Previous Reviewer 2)

This interesting article investigated the relationship between HCV infection and cardiovascular disease risk across a large population.

The work deals with a topical issue. The methodology is correct, and the conclusions are supported by the results. However, this reviewer raises some issues that need to be addressed by the authors.

1- A paragraph on the limitations of the study is missing at the end of the discussion.

2- In the discussion it is necessary to underline that recently it has been shown that sustained viral response (SVR) of HCV infection by direct antiviral drugs (DAAs) has important extra-hepatic favorable clinical effects, showing not only the correlation between HCV infection and vascular diseases but also the possibility of a direct reduction in CV risk and more. In fact, following SVR by DAAs, both a significant reduction in MACEs (Atherosclerosis Volume 296, Pages 40 - 47 March 2020. doi: 10.1016/j.atherosclerosis.2020.01.010 - Nutrition, Metabolism & Cardiovascular Diseases 2021; 31, 2345e2353. doi: 10.1016/j.numecd.2021.04.016) and a reduction in progression towards diabetes (Diabetes, Obesity and Metabolism Volume 22, Issue 12, Pages 2408-2416 December 2020. doi: 10.1111/dom.14168) have recently been observed. In addition to above-described extra-hepatic effects, SVR by DAAs leads to an important reduction in the risk of HCC (1- J Transl Med. 2019 Aug 28;17(1):292. doi: 10.1186/s12967-019-2033-x. 2- Cancers (Basel). 2020 May 26;12(6):1351. doi: 10.3390/cancers12061351). All these issues with the above references should added in the text as well as commented on in the discussion.

Author Response

Point 1:

1- A paragraph on the limitations of the study is missing at the end of the discussion.

Response 1:  Thank you very much for drawing attention to such an important aspect. I agree  and I am adding to the discussion in the presented work.

A limitation of the study was to some extent the cardiovascular disease analysis only based on patients' medical records, which might not contain all relevant data on cardiovascular disease and other relevant risk factors. Also, the assessment of liver fibrosis based on liver elastography, although it is one of the methods recommended and used routinely in hepatology, does not always reflect the actual stage of the disease. It mainly concerns advanced fibrosis at the F4 level according to Metavir, which does not always correlate with clinical cirrhosis. Also, the age distribution of the patients, although the mean age was 58 years, but the youngest patient was 19 years old, could have influenced the results obtained. With age, the risk of cardiovascular risk factors, cardiovascular disease and hypertension increases.

Point 2:

2- In the discussion it is necessary to underline that recently it has been shown that sustained viral response (SVR) of HCV infection by direct antiviral drugs (DAAs) has important extra-hepatic favorable clinical effects, showing not only the correlation between HCV infection and vascular diseases but also the possibility of a direct reduction in CV risk and more. In fact, following SVR by DAAs, both a significant reduction in MACEs (Atherosclerosis Volume 296, Pages 40 - 47 March 2020. doi: 10.1016/j.atherosclerosis.2020.01.010 - Nutrition, Metabolism & Cardiovascular Diseases 2021; 31, 2345e2353. doi: 10.1016/j.numecd.2021.04.016) and a reduction in progression towards diabetes (Diabetes, Obesity and Metabolism Volume 22, Issue 12, Pages 2408-2416 December 2020. doi: 10.1111/dom.14168) have recently been observed. In addition to above-described extra-hepatic effects, SVR by DAAs leads to an important reduction in the risk of HCC (1- J Transl Med. 2019 Aug 28;17(1):292. doi: 10.1186/s12967-019-2033-x. 2- Cancers (Basel). 2020 May 26;12(6):1351. doi: 10.3390/cancers12061351). All these issues with the above references should added in the text as well as commented on in the discussion.

Response 2:  

Thank you very much for your valuable suggestions. Yes, I agree with them. HCV eradication with DAA therapy reduces not only the risk of cardiovascular disease, but also other extrahepatic manifestations, including risk factors for cardiovascular disease, such as diabetes,  innsulin resistance and hepatic steatosis. But also reduced risk of progression of liver fibrosis to cirrhosis, hepatocellular carcinoma and of liver transplants. This will be added to the work.

Reviewer 2 Report (New Reviewer)

I read with great interest the paper “Hepatitis C Infection as A Risk Factor for Hypertension and Cardiovascular Diseases – EpiTer Multicentre Study" by Rajewski et al.

The article is well written. Paper design is fine. The article is logically divided into sections and subsections. Discussion is long, but it is just fine and well discussed in every point. Reference reported are relevant

Comments:

1)     introduction, line 79-89: nowadays we have the possibility to treat C Hepatitis through Direct Acting Antivirals. It has also been highlighted that virus eradication affect metabolic disorders with improved cardiovascular disease onset and progression please also add this point and it could also been added to the conclusion as a future perspective: treating everybody to improve outcomes (doi: 10.1111/dom.14168; doi: 10.1016/j.numecd.2021.04.016).

Author Response

Point 1:

1)     introduction, line 79-89: nowadays we have the possibility to treat C Hepatitis through Direct Acting Antivirals. It has also been highlighted that virus eradication affect metabolic disorders with improved cardiovascular disease onset and progression please also add this point and it could also been added to the conclusion as a future perspective: treating everybody to improve outcomes (doi: 10.1111/dom.14168; doi: 10.1016/j.numecd.2021.04.016).

Response 1:  

Thank you very much for your valuable suggestion. Of course, I will put it in the discussion, to highlight a very important aspect of treating HCV patients. Treatment with DAA affects practically all asepsis of extrahepatic HCV manifestations, including those that translate directly into the reduction of cardiovascular risk. Such as, for example, insulin resistance, diabetes, fatty liver, or carotid atherosclerosis quoted by your reviewer.

Reviewer 3 Report (New Reviewer)

In this paper, "Hepatitis C infection as a risk factor for hypertension and cardiovascular diseases – EpiTer Multicentre Study" by Rajewski et al., the authors intend to present the role of HCV infection as a risk factor for cardiovascular diseases and hypertension. The data comes from a multicentric antiviral treatment study, EpiTer, including 2898 adult patients with chronic HCV infection. The authors aimed to analyze the prevalence of cardiovascular diseases and hypertension in this cohort and to correlate it with age, liver fibrosis, and genotype. Unfortunately, even though the results seem interesting to the readers, the manuscript suffers from clearness, use of the English language, and some errors in the presentation of the results. Therefore, the authors must revise the paper extensively before being considered again for publication:

- correct the English language all over the paper

- rewrite the abstract to include results, exact data, and structure it (Background, aim, methods, results, conclusions)

- rewrite the Introduction (almost every sentence is in a separate paragraph)

- rewrite and improve the representation of the Methods (inclusion criteria; exclusion criteria; what cardiovascular diseases are considered, how are those diagnosed; how the hypertension was evaluated; how the liver fibrosis was evaluated - what device, when was evaluated; how the genotype was assessed - lab kit; what was the treatment, and so on).

- verify and improve the representation of the results: data from table 2 must be compared to the rest of the cohort and include these data also; same for table 4; verify data from table 6: % in the column for Hypertension for sex, fibrosis, genotype - added are not 100%; both for <60 and >60). Why do you present in table 6 patients grouped in under and over 60, but for logistic regression, there are under 40, 40-60, and over 60?

- the Discussion section must be improved 

- rewrite the Conclusions, preferably without numbering and repeating the same idea *see no. 7 and 8)

- the list of References must be edited according to the rules of the Journal for references.

Author Response

Response to Reviewer 3 Comments

Thank you very much for your valuable comments and suggestions. They will be included in the article.

Response

Inclusion criteria - each patient meeting the criteria for the treatment of chronic hepatitis  C under the Drug Program of the National Health Fund in Poland was included in the study. Patients over 18 years of age, diagnosed with HCV RNA over 6 months of age and liver fibrosis at least F1 according to Metavir or extrahepatic HCV manifestations,  regardless of the presence of liver fibrosis, were eligible for treatment. Exclusion criteria   - age under 18, pregnancy or breastfeeding, no confirmed HCV RNA in blood serum, no  evidence of fibrosis in liver biopsy or liver elastography, no patient consent.                                             

The presence of cardiovascular diseases and hypertension was assessed on the basis of the analysis of patients' medical records. Cardiovascular diseases that were taken into account in the presented study are ischemic heart disease in the form of - chronic coronary syndrome (stable angina pectoris), previous acute coronary syndrome (myocardial infarction), previous coronary angioplasty - PTCA, past coronary artery by pass grafting- CABG, previous ischemic stroke, previous transient ischemic attack - TIA, carotid atherosclerosis, lower limb atherosclerosis, previous lower limb artery angioplasty, aortic aneurysm. Liver fibrosis was assessed on the basis of classic core  needle biopsy or liver elastography performed with the FiboroScan Touch 502 device. The histopathological results of liver biopsies up to 5 years ago were taken into account,and the results of liver elastography up to 1 year back. The determination of HCV RNA  and HCV genotype was performed in standardized laboratories by PCR and nucleic acid   hybridization methods. Patients were treated with DAA under the Drug Program of  the National Health Fund in Poland. The type and duration of treatment depended on the HCV genotype, stage of fibrosis, history of prior treatment, and drug interactions. Treatment was done with: sofosbuvir, ledipasvir; sofosbuvir, ribavirin; elbasvir,grazoprevir; ombitasvir, paritaprevir, ritonavir, dasabuvir; ombitaswi, paritaprevi,  ritonavir; daclatasvir, asunaprevir. Combined with ribavirin in certain cases.

In the table on hypertension and cardiovascular diseases, the percentage data concerning the number of women, men, genotype type, and degree of fibrosis, or refer to the total number of women and men in the study group.

The grouping of patients initially into two groups under and over 60 years of age, and then for logistic regression into three groups - under 40, 40-60 and over 60 years of age, was dictated by the results I obtained when divided into 2 age groups and the desire to check the problem of the relationship between HCV genotype and arterial hypertension, hence additional analyzes and logistic regression division. The analysis in age groups showed that there is rather none the relationship between the occurrence of hypertension and gender (which was suggested by analysis of the whole group of patients) and the effect I observed was related to age and gender interaction, i.e. the women in this study were older, a hypertension is more common in the elderly and that is why the gender turned out to be here become important. Besides, when it comes to clarifying the mystery, is it relationship between genotype and hypertension - analysis in age groups showed that in the age group <40 and> 60 there is no relationship, however dependence was shown by the analysis in the group 40-60 and there genotype 3 was related however, with less frequent occurrence of hypertension. Such differences may arise from that in this age group there were the most patients with genotype 3 in the age group> 60, there were fewer of them.

Thank you very much for the suggestions on the applications, they are already revised and the references.

Round 2

Reviewer 1 Report (Previous Reviewer 2)

The authors have addressed enough of the issues raised by this reviewer.

Only minor comments remain.

References on the effects of DAAs on MACE, insulin-resistance/type 2 diabetes and HCC were included aggregated in the text ,and not separately after each new sentence added to the discussion.  Furthermore, some of the important references suggested by this reviewer strictly related to the issues addressed by the authors have not been added to the manuscript.

Author Response

Thank you once again for your valuable suggestions and for drawing your attention to such a significant problem. Highlighting the impact of HCV treatment on reducing MACE, insulin resistance and diabetes may contribute to a more active search for HCV patients and their prompt treatment with DAA.

Reviewer 3 Report (New Reviewer)

The manuscript still needs important improvements. Some recommendations were not followed and may be important to improve the manuscript. A native speaker must revise the English language. 

The abstract was not changed... too many "one..." (...is hearing, ...finds, ...found). "Results received?"  The abstract does not reflect any results of the study.

Still, the tables contain data that are probably not checked or presented in an incorrect form (if in a column the authors chose to put % of the total of that column, in the next one, the % is from something else probably).

Tables 1, 2, and 4 could be mixed to create a single one, to limit the number of tables, and probably to clarify why the % is not from the same reference group.

Rewrite the conclusions, and better not to use the numbering.

There are also typing errors that should be checked.

Author Response

Thank you for your valuable comments. Of course, I will edit the summary, conclusions, summary and conclusions as suggested, and the tables will provide an explanatory commentary on what the percentages in it relate to or edit it.

This manuscript is a resubmission of an earlier submission. The following is a list of the peer review reports and author responses from that submission.

Round 1

Reviewer 1 Report

This article has many comments. The relation between HCV infection and hypertension is limited as a comorbid condition rather than a causal relationship. 

Introduction:

Line 79: The recent years studies have proved that hepatitis C infection contributes also to the development of metabolic disorders.

Although this is right in general, however, it is related more to metabolic syndrome and insulin resistance but not hypertension. Of course this is more clear in Fatty liver disease or NASH. Again, the genotype most responsible for fatty liver infiltration in HCV infection and hence metabolic disturbance is genotype 3 that has low incidence in your country and hence I do not expect that the metabolic disturbance will be that problematic.

Line 86-88: The observed increased presence of heart and vessels diseases among patients suffering from chronic hepatitis C virus contributed to the indication of HCV as a new risk factor for cardiovascular diseases, and its presence as an extra-hepatic manifestation of hepatitis C infection.

You need to have reference and Iam not sure we have such relation.

Line 90-93: Cardiovascular diseases, including ischaemic heart disease, cerebrovascular diseases and peripheral vascular diseases are the reason for about 4,3 million of deaths in Europe annually which stands up for nearly half of all deaths – 48% (54% of women and 43% of men). In Poland the numbers are respectively 46% of deaths annually (51,1% of women and 40,9% of men).

What is the relation of HCV to this high mortality!!!!no relation.

Line 95-99: The incidence frequency of ischaemic heart disease increases together with the age, from 5–7% among women with the age range 45–64 to 10–12% among women with the age range 65–84, and 4–7% among men with the age range 45–64 to 12–14% among men with the age range 65–84 [12]. The main risk factors for the development of cardiovascular diseases in Poland are hypertension, lipid disturbances, obesity, diabetes, and smoking.

Again you are discussing ischemic heart disease with HCV has no relation to!!!

Line 99: The main risk factors for the development of cardiovascular diseases in Poland are hypertension, lipid disturbances, obesity, diabetes, and smoking.

This is the general risk factors all over the world!!! what is new!!!

Line 102: In the last years the infection of HCV seem to be indicated as a possible developmental reason for cardiovascular diseases [13,14].

The reference you use indicate that mainly this may be true if the hypertension is combined with diabetes mellitus.

Line 107-108: hepatitis C infection leads in indirect way to the development of other key risk factors – diabetes, obesity, hypertriglyceridemia, hypertension and chronic kidney diseases.

HCV infection associated with insulin resistance and may be a risk factor for DM only; but not other disease. The obesity, hypertriglyceridemia and CRD are all risk factors for hypertension and not the reverse!!!

Line 115-116: The most vital seem to be carbohydrate disorders which result in the development and increase of arteriosclerosis both of coronary arteries, as well as, brain arteries and the peripheral ones. 

This needs reference.

Material and method:

Line 135: The summary statistics for continuous, non-normally distributed variables are pre..

Statistical analysis has to be written under separate title.

What is the type of this study??? Is it randomized?? cohort?? Prospective or retrospective?? How you select your patients? What are the inclusion and exclusion criteria?? This is bias!!!. Is there is control group with HCV infection and no hypertension?? Why you do not make a comparison between hypertensive and normal blood pressure patients in your study group? Why there is no HCV-RNA-PCR results included??

Table (1): you mention fibrosis either by liver biopsy or fibroscan. Fibrosis graded as 1,2,3 and 4. What is the classification you used??!! Is it histopathology and which classification??!! Why 154 degree of liver fibrosis did not determined???!! Also table (1) should be included under the result section.

Table (2): 1132 out of 2898 of your included patients had hypertension !!!!!!!!!!!!!! this simply means that around 40% are affected. Very difficult to accept this numbers but it looks to me bias in selection!!! If this is true, then you better look to the problem of hypertension itself.

Table (2 and 3): You find more hypertension in patients with F3,4 more than F1,2. You know that the more the fibrosis degree up to F4; full cirrhosis, there is more severe portal hypertension and consequently more portosystemic shunts to open. In other wards, hypotension rather than hypertension is noticed and expected. How you can explain???!!!

Another point in table 3: Line 159: Patients infected by HCV genotype 1 or 4 suffered from hypertension more often than patients with HCV genotype 3.

What is your explanation??!!! It is known that genotype 3 which is rare in your studied group; is the most known type to cause hepatic steatosis and consequently the most expected to produce metabolic disturbance!!!!

Table (4): Line 161-162:The incidence of cardiovascular disease in the studied group of patients was as follows- all incidents – 131 (4,52%), including :Ischaemic heart disease 104 (3,95%), stroke 2 (0,07%), atherosclerosis 21 (0,72%), aneurysms 4 (0,14%).

How can you determine atherosclerosis???

Table (6): ????!!!!!!!!!!!

Discussion: 

Line 184: The studies of the last years have proved the relevance of chronic hepatitis C infection...

You give only one reference and say the studies???! Previously, i explain the problem of this reference. Do you have other references???

Line 193-194: The number decrease of acute coronary syndromes and ischaemic strokes has been proved, diseases of patients suffering from chronic hepatitis C virus and treated casually in comparison to patients who were not treated [18].

This reference is reporting the peripheral vascular diseases and not coronaries!!!

Line 206-207: Cardiovascular diseases were found in 4,55% of patients suffering from HCV. The incidence was characterized by a similar frequency both of women and men, which constitutes a similar percentage in relation to the general population in Poland.

This is what Iam saying!!! However, you also need to find explanation of your contradictory results of the female sex preponderance over male.

Line 250-251: Extra-hepatic manifestations of HCV such as glomerulonephritis – membranous and membrano-proliferate ones, which may be the reason for hypertension in the study group, but have not been analyzed in the given study, could also contribute to the received result.

It is mandatory to know if there is renal affection or not because this should be one of the exclusion criteria.

Line 258-263:  In the studied population of patients with HCV, hepatic fibrosis was the most often assessed by means of non- invasive methods – mainly by means of elastography of a liver with fibroscan method, which may explain the fact that there were no patients with ascites and clinical features of hepatic cirrhosis (B and C class according to Child-Pugh scale) among patients with advanced fibrosis of F3 and F4 type, and the majority of them was characterized by A class according to Child-Pugh scale [22–25]. 

Controversy!!!! Whatever method you use F3,4 mean liver cirrhosis!!!!!! with or without ascites; i.e vasodilatation and hypotension and not hypertension. You mension that most of your patients are Child A and those patients never have F3,4!!!!!!!!!!

Conclusion:

I completely disagree with your conclusions as it is relay on bias in selection and bad study design.

Reviewer 2 Report

This manuscript is interesting. However, this reviewer raises some issues that need to be addressed by the authors.

Major comments

1- At the end of the discussion a paragraph is needed that addresses the limitations of the study.

2- The authors, in their conclusions, state that "The role of Hepatitis C infection as a risk factor for cardiovascular diseases needs further studying. " Instead, a close correlation between major cardiovascular events (MACEs) and HCV infection has recently been confirmed by the protective role of the eradication of C virus infection on CV outcomes. In fact, following a sustained virologic response by direct-acting antivirals (DAAs), a significant reduction in MACE has recently been observed (Atherosclerosis Volume 296, Pages 40 - 47 March 2020. doi: 10.1016/j.atherosclerosis.2020.01.010 - Nutrition, Metabolism & Cardiovascular Diseases 2021; 31, 2345e2353. doi: 10.1016/j.numecd.2021.04.016). This important issue, which indirectly supports the hypothesis formulated by the authors, and the above references should be commented in the text.

3- Arterial hypertension is due to insulin resistance and is therefore closely related pathophysiologically to type 2 diabetes in the Metabolic Syndrome. Interestingly, the eradication of HCV infection with DAA, leading to a reduction in chronic inflammatory status, results in a reduction in the incidence of type 2 diabetes (Diabetes Obes Metab. 2020 Dec; 22(12):2408-2416. doi: 10.1111/dom.14168.). This issue should also be addressed in discussion.

Minor comments

1- When writing numbers in thousands, it needs a comma between hundreds and thousands (for example 1,000).

2- A linguistic revision by a native English speaker is required.